# Effects of the N-Butanol Extract of Pulsatilla Decoction on Neutrophils in a Mouse Model of Ulcerative Colitis

**DOI:** 10.3390/ph17081077

**Published:** 2024-08-16

**Authors:** Yadong Wang, Hui Wu, Juan Sun, Can Li, Ying Fang, Gaoxiang Shi, Kelong Ma, Daqiang Wu, Jing Shao, Hang Song, Tianming Wang, Changzhong Wang

**Affiliations:** 1Department of Pathogenic Biology and Immunology, College of Integrated Chinese and Western Medicine (College of Life Science), Anhui University of Chinese Medicine, Hefei 230012, China; wangyadong4925@sina.com (Y.W.); wuhui2474170599@outlook.com (H.W.); sunjuanahucm@163.com (J.S.); deerlyee@hotmail.com (C.L.); fangying2023ahtcm@163.com (Y.F.); ssjlsgx@163.com (G.S.); makelong210@126.com (K.M.); daqwu@126.com (D.W.); ustcnjnusjtu@126.com (J.S.); hangsong@ahtcm.edu.cn (H.S.); wtm1818@163.com (T.W.); 2Institute of Integrated Traditional Chinese and Western Medicine, Anhui University of Chinese Medicine, Hefei 230012, China; 3Anhui Province Key Laboratory of Chinese Medicinal Formula, Hefei 230012, China

**Keywords:** ulcerative colitis, n-butanol extract of Pulsatilla decoction, neutrophils

## Abstract

Ulcerative colitis (UC) is a chronic inflammatory disease, the incidence of which is increasing worldwide. However, the etiology and pathogenesis of UC remains unclear. The n-butanol extract of Pulsatilla decoction (BEPD), a traditional Chinese medicine, has been shown to be effective in treating UC. This study aimed to explore the molecular mechanism underlying the effects of BEPD on UC, in particular its effects on neutrophil extracellular trap (NET) formation by neutrophils. High-performance liquid chromatography was used to determine the principal compounds of BEPD. UC was induced in mice using dextran sodium sulfate, and mice were treated with 20, 40, or 80 mg/kg BEPD daily for seven days. Colonic inflammation was determined by assessing the disease activity index, histopathology, colonic mucosal damage index, colonic mucosal permeability, and pro- and anti-inflammatory cytokine levels. The infiltration and activation status of neutrophils in the colon were determined by analyzing the levels of chemokine (C-X-C motif) ligand (CXCL) 1 and CXCL2, reactive oxygen species, Ly6G, and numerous NET proteins. The findings suggest that BEPD improved the disease activity index, histopathology, and colonic mucosal damage index scores of mice with UC, and restored colonic mucosal permeability compared with untreated mice. The expression levels of the pro-inflammatory cytokines interleukin-1β, interleukin-6, and tumor necrosis factor-α in colon tissues were significantly decreased, while the expression levels of anti-inflammatory cytokines in colon tissues were significantly increased, exceeding those of control mice. In addition, BEPD reduced the expression of the neutrophil chemokines CXCL1 and CXCL2 in the colon tissue of mice with UC, reduced neutrophil infiltration, reduced reactive oxygen species levels, and significantly reduced the expression of NET proteins. BEPD also significantly reduced NET formation. The results of this study suggest that BEPD exerts therapeutic effects in a murine model of UC by inhibiting neutrophil infiltration and activation in the colon, as well as by inhibiting the expression of key proteins involved in NET formation and reducing NET formation, thereby alleviating local tissue damage and disease manifestations.

## 1. Introduction

Ulcerative colitis (UC) is a chronic inflammatory disease affecting the colon [1,2]. It manifests clinically as ulcerative lesions in the colon [3], with mucosal and submucosal edema and continuous infiltration of inflammatory cells [4]. Lesions typically develop in the sigmoid colon and gradually disseminate into the descending, transverse, and ascending colon as the condition progresses. Severe cases can involve the entire colon in a continuous distribution [1,5]. Symptoms include weight loss, changes in bowel habits, recurrent bloody diarrhea, and abdominal pain [6]. Epidemiological data have revealed that UC is prevalent in developed regions, such as Europe and the United States, and that its incidence in China is increasing annually. Although the mortality rate of UC is low, long-term chronic inflammation increases the risk of colorectal cancer, other gastrointestinal diseases, infections, and respiratory diseases [7,8,9]. The etiology and pathogenesis of UC remains unclear, resulting in a lack of treatment options, although newly developed drugs have brought relief to some patients with UC. Studying the pathogenesis of UC may provide a theoretical basis for the development of novel therapeutic strategies.

Damage to the colonic mucosa results in the immediate recruitment and activation of polymorphonuclear neutrophils (PMNs). Activated PMNs secrete reactive oxygen species (ROS) and inflammatory factors, inducing further tissue damage [10], which often presents as acute ulcers. The average lifespan of PMNs following tissue infiltration is 24–48 h. However, production of inflammatory factors such as tumor necrosis factor (TNF)-α, interleukin (IL)-8, and IL-1 in the colon leads to the continuous recruitment and activation of PMNs. This results in the main pathological feature of UC: chronic ulcerative damage to the colonic mucosa. Previous studies have confirmed the uncontrolled accumulation and persistence of PMNs in damaged mucosal lamina propria [11]. A vicious cycle is formed, in which infiltrating PMNs damage the mucosa, which in turn leads to further PMN infiltration. This is the foundation of the development of clinicopathological changes in patients with UC. Therefore, some scholars regard the degree of neutrophil infiltration in the lamina propria of the intestinal mucosa, the degree of accumulation in the crypt, the destruction of the crypt, and the formation of crypt abscesses as morphological markers and severity indicators of active UC [12,13,14,15].

PMNs constitute the first line of immune defense, with their primary role involving phagocytosis of pathogenic microorganisms. Inside PMNs, phagosomes are fused with intracellular lysosomes to form phagocytic lysosomes, which kill and degrade encapsulated pathogens using lysosomal enzymes. This method does not result in tissue damage; however, it is limited by the number of PMNs available. PMNs can also kill pathogenic microorganisms by releasing antimicrobial factors, including granular enzymes, ROS, and neutrophil elastase (ELANE). Cui et al. [16] showed that PMNs release a large amount of granular proteins and ELANE into the extracellular space, inducing tumor cell apoptosis. ELANE exists only in neutrophils and can hydrolyze fibronectin, proteoglycans, type IV collagen, and extracellular matrix proteins, causing tissue damage, prolonging the inflammatory response time, and preventing wound healing. However, the various enzymes released are rapidly diluted by the large amount of exudate in the inflamed area, decreasing their effectiveness, and they subsequently undergo rapid and complete neutralization by anti-proteases. Neither phagocytosis nor the secretion of antimicrobial factors can explain the relationship between infiltrating PMNs and inflammation. However, another PMN mechanism of action has been identified. PMNs release neutrophil extracellular traps (NETs), a DNA network fiber structure that combines various proteins such as histones, ELANE, and myeloperoxidase (MPO), maintaining them at high concentrations at the inflammatory site to resist neutralization. The DNA network fiber structure also entraps pathogenic microorganisms, exerting a powerful killing effect but also causing severe tissue damage, playing a key role in the development of UC [17,18,19,20].

Traditional Chinese medicine believes that the etiology and pathogenesis of ulcerative colitis is the weakness of spleen and stomach, which leads to damp-heat obstructing the intestinal tract, blocking qi machinery, and damaging the intestinal collaterals [21]. Pulsatilla decoction (PD) is a Traditional Chinese medicine formulation recorded in Shang Han Lun, a classic work of Chinese medicine written by Zhang Zhongjing in the Eastern Han Dynasty. Each dose of Pulsatilla decoction consists of Pulsatilla Chinensis (Bunge) Regel 15 g, Phellodendron chinense C.K. Schneid 12 g, Coptis chinensis Franch 6 g, Fraxinus rhynchophylla Hance 12 g, and is boiled with water to obtain the decoction. It holds the function of clearing away heat, detoxifying cooling blood, drying dampness, and stopping dysentery [22]. PD is widely used in the treatment of UC [23,24]. Modern pharmacological studies have found that the main active ingredients of PD include anemoside B4, berberine, phellodendrine, jatrorrhizine, epiberberine, palmatine, magnoflorine, berberrubine, esculin, esculetin, fraxin and fraxetin, etc., and functioning anti-inflammatory, anti-viral, anti-oxidation, anti-tumor, anti-ulcer, antibiotics and antifungal activity. We have previously demonstrated that BEPD exerts a therapeutic effect on UC by inhibiting PMN infiltration and the release of pro-inflammatory cytokines [25,26]. In this study, the impact of BEPD on PMNs and NETs were investigated, to determine whether this mechanism underlies the therapeutic effect of BEPD on UC.

## 2. Results

### 2.1. Main Components of BEPD

To identify the main active compounds in BEPD, high-performance liquid chromatography (HPLC) was used to examined BEPD (Figure 1). Anemoside B4, phellodendrine, esculin, esculetin, epiberberine, berberine, and jatrorrhizine were selected for identification based on their respective BEPD criteria. The retention times, peak areas, and contents are listed in Table 1. Among these constituents, anemoside B4, berberine, and esuclin were the most abundant.

### 2.2. BEPD Significantly Improves the General Condition and Disease Activity Index (DAI) Scores of Mice with UC

The UC mouse model was successfully established. Compared with control mice, model group mice began to lose weight on day 4 (Figure 2A), developing loose and then bloody stools. These changes occurred alongside increased DAI scores (*p* < 0.05) (Figure 2B). From day 9, the trend of weight loss slowed, consistent with the self-healing characteristics of DSS-induced UC [27], but weight loss continued until the end of the experiment. The weight of all treated mice began to increase and the DAI score began to decrease from the second day of treatment. By the third day of treatment, there was a significant difference between the treatment and model groups (*p* < 0.05). At the end of the experiment, the DAI score in the high-dose BEPD (BEPD-H) group was significantly lower than that in the medium-dose BEPD (BEPD-M), low-dose BEPD (BEPD-L), and mesalazine (Mes) groups, indicating superior treatment efficacy (*p* < 0.05). The effect in the Mes group was greater than that in the BEPD-M and BEPD-L groups.

### 2.3. Impact of BEPD on UC Lesions

UC is accompanied by the formation of chronic ulcers in the colonic mucosa, as well as changes in colon length and histopathological and colonic mucosa damage index (CMDI) scores. Compared with control mice, model mice had significantly shorter colons (*p* < 0.01). After one week of treatment, the colon length of the Mes and BEPD-L groups did not differ significantly compared with the model group, whereas the BEPD-H and BEPD-M groups showed a significant recovery in colon length (*p* < 0.05), with the greatest effect in the BEPD-H group (Figure 3A,B).

HE staining of colon tissue revealed that compared to control group mice, model group mice exhibited obvious signs of colonic mucosal edema, erosion, ulceration, mucosal layer damage, and inflammatory cell infiltration (Figure 3C), with higher histopathological (Figure 3D) and CMDI scores (*p* < 0.01) (Figure 3E). Colonic mucosal edema, erosion, ulceration, mucosal layer damage, and inflammatory cell infiltration were reduced in all treated mice compared with model mice, and histopathological and CMDI scores decreased. However, there were no significant differences between the Mes and model groups, whereas there were significant differences between all BEPD treatment groups and the model group.

Colonic mucosal permeability was significantly increased in model group mice, consistent with the mucosal damage observed with hematoxylin-eosin (HE) staining (Figure 3F). Mice treated with Mes (*p* < 0.05) or BEPD (*p* < 0.01) showed a decrease in colonic mucosal permeability, with the effect of BEPD significantly greater than that of Mes (*p* < 0.05). BEPD exhibited a dose-dependent effect, with BEPD-H having the greatest impact.

### 2.4. Effects of BEPD on Cytokine Production

ELISA results showed that the levels of pro-inflammatory cytokines (IL-1β, IL-6, and TNF-α) were significantly increased in the colon tissues and serum of model mice, whereas the levels of anti-inflammatory cytokines (IL-4 and IL-10) were decreased, consistent with the disease manifestations of UC. In mice treated with Mes, IL-1β levels in the colon tissues and serum (*p* < 0.01), IL-6 levels in the serum, and TNF-α levels in colon tissues decreased (*p* < 0.05), while IL-10 levels decreased (*p* < 0.05) and IL-4 levels increased (*p* < 0.05). This suggests that Mes mainly exerts its therapeutic effect on UC through effects on IL-1β and IL-4. BEPD-H group mice showed a significant decreases in the levels of IL-1β, IL-6, and TNF-α in the colon tissues (*p* < 0.05), which returned to the levels seen in control mice. Serum TNF-α levels did not differ significantly between BEPD and model group mice. However, anti-inflammatory cytokine levels in colon tissues increased significantly in BEPD group mice, exceeding the levels seen in control mice (*p* < 0.05) (Figure 4). This indicates that BEPD plays a therapeutic role in UC by inhibiting the release of pro-inflammatory cytokines and promoting the release of anti-inflammatory cytokines in colon tissues to alleviate the inflammatory response.

### 2.5. Effects of BEPD on Chemokine Production and PMN Infiltration and Activation

Since PMNs play an important role in the development of UC, it has been speculated that BEPD has therapeutic effects on PMNs in UC. Levels of the PMN chemokines (C-X-C motif) ligand (CXCL)1 and CXCL2 were decreased in colon tissues of all treated mice compared with model mice, especially those treated with high- and medium-dose BEPD (*p* < 0.01) (Figure 5A).

Immunofluorescence staining of Ly6G, a PMN-specific marker, revealed large numbers of infiltrating PMNs in the colonic mucosa of model mice (*p* < 0.01), which were reduced to varying degrees by treatment (*p* < 0.01) (Figure 5B,C). BEPD treatment showed a certain dose-dependent manner; BEPD-H was more effective at reducing PMN infiltration than Mes (*p* < 0.05).

PMN activation was evaluated by staining for ROS. ROS cytoplasmic staining significantly increased in the colon tissues of model mice (*p* < 0.01), and decreased in all treated mice (*p* < 0.05), with BEPD-H treatment showing the most significant effect (*p* < 0.01), and BEPD-M and BEPD-L treatment showing similar effects to Mes (Figure 5D,E). These results indicate that the effects of BEPD on UC are related to the infiltration and activation of PMNs.

### 2.6. Effects of BEPD on the Formation and Release of NETs

Western blotting (WB) showed that compared with control mice, the expression levels of key proteins involved in NET formation (PAD4, ELANE, MPO, and Cit H3) were significantly increased in the colonic mucosa of model mice (*p* < 0.01). Treatment partially reversed these changes. In Mes group mice, the expression levels of PAD4, MPO, and Cit H3 decreased to varying degrees compared with those in model mice (*p* < 0.05), but remained significantly higher than those in control mice (*p* < 0.01). Mes showed no effect on the levels of ELANE. This indicates that although Mes can reduce the transcriptional and translational activities of ELANE, MPO, and PAD4 in model mice, chromatin decondensation is still active. Mice treated with high-dose BEPD showed a significant decrease in the expression levels of ELANE (*p* < 0.05), PAD4 (*p* < 0.01), MPO (*p* < 0.01), and Cit H3 (*p* < 0.01) compared with model and Mes group mice. ELANE and Cit H3 expression levels in the low-dose group were similar to those in the Mes treatment group, and in the medium-dose group were intermediate between those in the high- and low-dose groups; the expression levels of PAD4 and MPO did not differ from those in model mice (Figure 6).

Immunohistochemistry showed that staining of these key proteins increased significantly in model mice, as did staining of the NET-associated components MMP-8 and -9 (*p* < 0.01). MMP-8 and -9 were expressed in the cytoplasm, while the key proteins involved in NET formation, such as PAD4, ELANE, and MPO were expressed in both the cytoplasm and the nucleus (Figure 7A–E), indicating the involvement of these proteins in NET formation. Compared with model mice, staining of MMP-8, MMP-9, PAD4, ELANE, and MPO decreased following BEPD-H treatment (*p* < 0.01). The high-dose group showed the most significant effect, indicating a dose-dependent response.

After NET formation, these proteins are released into the extracellular space and enter the intestinal cavity as a result of mucosal damage, eventually being excreted from the body in the feces. Therefore, fecal samples were analyzed by using ELISA to detect NET-related proteins. Compared with control mice, the protein expression levels of MPO, ELANE, and MMP-8 and -9 increased significantly in the feces of model mice (*p* < 0.05) (Figure 7F). The protein expression levels of ELANE and MPO decreased in Mes group mice compared with model mice (*p* < 0.05), while the expression level of MMP-9 increased slightly (*p* < 0.05). There was no difference in MMP-8 expression levels. This suggests that treatment of UC with Mes does not greatly impact NET formation. BEPD group mice showed significantly decreased protein expression levels of MPO (*p* < 0.05), ELANE (*p* < 0.01), and MMP-8 (*p* < 0.05) and -9 (*p* < 0.01), compared with those in model and Mes group mice, with the high-dose group exhibiting the greatest effects. This indicates that BEPD may alleviate UC through effects on NET formation.

The transcription and translation of ELANE, MPO, and PAD4, the key proteins of NET formation, are enhanced. PAD4 is activated and incorporated into the nucleus, in response to ROS, which degrades nucleosome histones and uncoils chromatin. Therefore, this study speculated that there may be a correlation between the expression of ROS and Cit H3 during NET formation. A Pearson correlation analysis between the WB analysis data of Cit H3 and the immunohistochemical quantitative data of ROS showed a strong correlation between ROS and Cit H3 (rs = 0.989, *p* < 0.001).

### 2.7. Effect of BEPD on Extracellular NET

Immunofluorescence co-localization analysis was used to demonstrate the formation of extracellular NET. By comparing Pearson’s and overlap, differences in NET formation in the colon tissues of mice were observed. MPO and Cit H3 co-located with DNA is a NET marker (Figure 8A); significantly increased NET formation was observed in the colon tissues of model mice compared with control mice (*p* < 0.01). Compared with model mice, Mes group mice showed a decrease in NET formation (*p* < 0.05), which was still significantly greater than that in control mice (*p* < 0.01), consistent with the decrease in PMN infiltration after treatment; BEPD group mice showed a dose-dependent decrease in NET formation (*p* < 0.01) (Figure 8B). As ELANE and MPO co-located with DNA (Figure 8C), significantly increased NET formation was observed in the colon tissues of model mice compared with control mice (*p* < 0.01). Compared with model mice, Mes group mice showed a decrease in NET formation (*p* < 0.01), which was still significantly greater than that in control mice (*p* < 0.01); BEPD group mice showed a dose-dependent decrease in NET formation (*p* < 0.01) (Figure 8D). Combined with the changes in PMN infiltration in the colonic mucosal tissue, this indicates that BEPD can inhibit extracellular NET formation.

## 3. Discussion

UC is a chronic intestinal inflammation characterized by colonic damage. Once it develops, it becomes a lifelong disease. The clinical incidence of UC is increasing [6], leading to a rapid increase in the number of patients. Consequently, the demand for UC treatment is escalating, resulting in the introduction of new drugs to help some patients maintain disease remission under drug control. However, long-term drug treatment for chronic diseases can lead to numerous adverse reactions. Aminosalicylic acid drugs [28] are the primary treatment for UC. Although the efficacy and safety of current clinical treatment drugs have improved, some patients remain insensitive to them [29,30], while others may experience severe adverse reactions [31]. Glucocorticoids (GCS) are also commonly used in UC treatment, but studies have shown that their use can disrupt the intestinal flora, increase the risk of intestinal flora imbalance, and cause drug resistance. Due to their significant side effects, they can only be used for short periods [32]. Immunosuppression has rapidly developed over the past 20 years, but it has been observed in clinical practice that many patients are not resistant to the drugs, leading to drug withdrawal due to adverse reactions [33]. Biological preparations have emerged as a new treatment for UC in recent years, but their high cost limits their widespread use [34].

In the present study, BEPD was used to investigate its possible mechanism of action. The mechanism of action of anti-UC drugs has been widely described in the existing literature, and studies have shown that their pharmacological mechanisms may involve maintaining the homeostasis and diversity of intestinal flora, increasing short chain fatty acid content and repairing the colonic mucosal barrier [35], but the mechanism of BEPD against UC is still unclear. At present, there is no literature report on the effect of BEPD on PMNs, and the formation and release of NETs in colon tissue. Therefore, this study will elaborate the effects of BEPD on the regulation of PMNs in colon tissue, the repair of colon tissue, and the formation and release of NETs, in order to reveal the pharmacological mechanism of BEPD.

The present study investigated the efficacy and underlying mechanisms of BEPD in treating mice with UC, and the findings demonstrated that BEPD exerts a protective effect on the colonic mucosa of UC mice by inhibiting the recruitment and chemotaxis of PMNs in the intestinal lamina propria. Mucosal epithelial injury, crypt abscess formation [18,36], diarrhea, hematochezia, weight loss [37], and other manifestations are typical pathological changes observed in UC mice. This study demonstrates that BEPD can effectively improve these symptoms by promoting restoration of the colonic mucosa. PMNs play a crucial role in the development and incidence of UC [38] as they are key inflammatory cells involved in the body’s defense reaction [18]. During UC progression, there is abnormal infiltration and accumulation of PMNs [19,20,39] in the lamina propria of intestinal mucosa along with increased activity levels [14]. The results indicate that BEPD effectively reduces DAI score by modulating PMN activity. Histological examination using HE staining revealed significant pathological changes in colonic mucosa from model group mice including edema, erosion, ulceration, disruption of the mucosal layer, and infiltration of inflammatory cells compared to normal group mice. However, treatment with medium- and high-dose BEPD restored the integrity of the mucosal layer while significantly reducing inflammatory infiltration compared to the low-dose BEPD group or the Mes group. These results indirectly indicated that BEPD improves overall condition, DAI score, and mitigates the inflammatory response in UC mice.

UC is primarily characterized by chronic inflammation in the intestines. During the disease progression, a substantial release of pro-inflammatory cytokines (such as TNF-α, IL-1β, and IL-6) occurs in the colon tissue, while there is a reduction in the secretion of anti-inflammatory factors (such as IL-4 and IL-10). Additionally, numerous inflammatory cells actively participate in local inflammatory responses. Among these cells, IL-1 β is produced and secreted by various cell types and typically plays a crucial role in host defense mechanisms. It is associated with pain perception, inflammation regulation, and autoimmunity processes [40]. Furthermore, IL-1 β also contributes to neuroprotection as well as tissue remodeling and repair mechanisms [41]. On the other hand, IL-4 can effectively mitigate pathological inflammation by promoting wound healing and fibrosis reduction [42]. Neutrophils serve as the body’s first line of cellular defense against pathogens and are among the initial cells recruited to sites of inflammation [43]. Subsequently activated locally recruited PMN play defensive roles such as phagocytosis, immunity and cause tissue damage. Activated PMNs continuously release ROS, ELANE, MMP, and MPO [39,44] in the intestinal mucosa, significantly enhancing tissue damage while losing their protective effect and contributing to tissue injury scenarios. This study demonstrated a significant increase in pro-inflammatory cytokines IL-1β, IL-6, and TNF-α, inhibition of anti-inflammatory cytokines IL-4 and IL-10, as well as a significant increase in chemokines CXCL1 and CXCL2 in the colon tissues of mice with UC. The If and IHC results showed increased expression of Ly6G and ROS in the colon tissues of UC mice, indicating massive recruitment and activation of PMNs. Additionally, histopathological score, CMDI score, and mucosal permeability were significantly increased in mice with UC-induced intestinal mucosal tissue damage. After treatment with BEPD, levels of pro-inflammatory cytokines decreased significantly while anti-inflammatory cytokines increased significantly in the colon tissues of mice. This further confirmed the anti-inflammatory function of BEPD. The expression levels of chemokines CXCL1 and CXCL2 decreased significantly along with reduced expression levels of Ly6G and ROS compared to the model group. These findings indicate that BEPD effectively limits infiltration and activation of neutrophils within colon tissues. Notably, high-dose BEPD exhibited more pronounced effects than the Mes group. The levels of IL-1β were decreased, while those of IL-4 were increased in all treatment groups. Notably, the BEPD treatment group exhibited a superior effect compared to the Mes group, suggesting that BEPD promotes colonic mucosal repair more effectively. These findings were consistent with alterations observed in histopathological score, CMDI score, and mucosal permeability in mice. The findings suggest that BEPD exerts a protective role by suppressing neutrophil chemotaxis and activation, facilitating tissue repair, thereby enhancing the efficacy of UC treatment.

To gain further insights into the effects of BEPD on NET formation and release, this study employed WB, IHC, and fecal samples ELISA to detect the expression of proteins related to NET formation. The results demonstrated increased expression of MPO, PAD4, ELANE, Cit H3, MMP-8, and MMP-9 in UC mice colon tissue indicating enhanced PMN respiratory burst, DNA unwinding, nucleosome degradation, and neutrophil granule protein release leading to extensive NET formation and release. However, after BEPD treatment, there was a decrease in the expression of these proteins with significant efficacy observed in the high-dose group compared to the Mes group. These findings suggest that BEPD can effectively inhibit NET formation and release while protecting colon tissue. Furthermore, immunofluorescence colocalization was utilized to confirm extracellular NET formation in this study which revealed substantial net release in UC mice colon tissue. Following BEPD treatment, there was a dose-dependent reduction in NET formation and release particularly evident in the high-dose group surpassing the effect seen with the Mes group. These results indicate that BEPD can effectively inhibit the formation and release of NET in colonic tissue of UC mice and reduce tissue damage.

## 4. Materials and Methods

### 4.1. Experimental Animals

A total of 72 healthy specific pathogen-free female Kunming mice (aged 6–8 weeks, weighing 18–20 g) were purchased from Jiangsu Huachuang Xinnuo Pharmaceutical Technology Co., Ltd. (Taizhou, China; license number SCXK [Su] 2020–0009). The mice were housed in a disease-free environment, with a humidity of 50–55%, a temperature of 22 ± 2 °C, a 12 h light/dark cycle, and free access to food and water. The animal experiments were approved by the Experimental Animal Ethics Committee of Anhui University of Traditional Chinese Medicine (permit number 2023007) and were conducted in accordance with Chinese legislation on the ethical use and care of laboratory animals.

### 4.2. Qualitative Analysis of BEPD Extract via HPLC

BEPD was prepared as previously described [44]. Radix pulsatillae, Cortex phellodendri, Rhizoma coptidis and Cortex fraxini were mixed at a ratio of 15:12:6:12, soaked in 80% ethanol, and heated in a 70 °C water bath overnight to extract the Pulsatilla decoction, refluxed evaporation for 3 h and filter after cooling. The process was repeated three times. The filtrate was combined, and then extracted with petroleum ether, ethyl acetate, and n-butanol to extract different effective parts separately. The extract was dried and evaporated at 80 °C by rotary evaporator, and finally, the BEPD was obtained. To identify the main active compounds in BEPD, we selected esuclin, aescin, epiberberine, berberine, phellodendrine, japonin, and anisidin B4 for analysis. High-performance liquid chromatography (HPLC) was performed to determine the main active compounds in BEPD. A sample of BEPD powder weighing 0.2 g was precisely weighed and passed through No. 2 sieve (850 μm ± 29 μm 24 mesh) and placed in a conical flask with stopper, and a mixed solution of methanol-hydrochloric acid (100:1) totaling 50 mL was accurately added. The bottle was sealed, weighed, and subjected to ultrasonic treatment (Power: 250 W, Frequency: 40 kHz) for 30 min. The bottle was then cooled, reweighed, and brought up to the original weight using methanol. The resulting mixture was shaken well, filtered, and 2 mL of the filtrate was precisely measured and placed in a 10 mL volumetric flask. Methanol was added into the volumetric flask for constant volume, shaken well, filtered, filtrate was taken. Liquid chromatography conditions: Syncronic C18, Dim 250 mm × 4.6 mm, 5 μm chromatographic column was used. Octadecyl silane bonded silica as filler, with acetonitrile (0.05 mol/L):potassium dihydrogen phosphate solution (50:50) (Added 0.4 g sodium dodecyl sulfate to each 100 mL mobile phase, and adjusted the pH value to 4.0 with phosphoric acid) as mobile phase, and the flow rate was 1 mL/min. The evaluation was conducted using an external standard quantitative method.

### 4.3. Establishment and Treatment of the UC Model

The 72 female Kunming mice were randomly divided into six groups, using the method developed by Ma et al. [38,39]. UC was induced in all mice except those in the control group by replacing drinking water with 3.0% dextran sodium sulfate (DSS) aqueous solution for seven days. Mice in the control group drank distilled water for seven days. After UC induction, the 60 mice with UC were divided into five groups according to body weight: model, Mes, BEP-H, BEPD-M, and BEPD-L. Mice in the control and model groups were administered with 0.5 mL·kg^−1^ normal saline, mice in the Mes group were administered with 200 mg·kg^−1^ Mes [45], and mice in the BEPD-H, BEPD-M, and BEPD-L groups were administered with 80, 40, and 20 mg·kg^−1^ BEPD, respectively. Mes and BEPD were prepared with normal saline; all solutions were administered by gavage once daily for seven days.

### 4.4. General Condition of Mice and Disease Activity Index Scores

The general health of mice, including their mental state, diet, activity, and fur condition, was monitored every day. Daily food intake, body mass, bowel movements, fecal blood, and mortality were recorded for each group. The DAI [46] score was calculated based on the criteria listed in Table 2.

### 4.5. Colonic Mucosal Permeability Detection

After a 12-h fast, 200 mL fluorescein isothiocyanate-dextran (FITC-dextran; 500 mg/kg; Sigma-Aldrich, St. Louis, MO, USA) was administered to each mouse by oral gavage. Mice were euthanized 4 h later and venous blood was collected. Serum was obtained through centrifugation and its absorbance was measured using a fluorescence spectrophotometer (excitation wavelength, 490 nm; emission wavelength, 520 nm). The serum concentration of FITC-dextran was calculated using a standard curve.

### 4.6. Histopathology

After mice were euthanized and blood was collected, a segment of colon approximately 0.5 cm long was cut 1 cm from the anus and immediately immersed in a 4% paraformaldehyde solution for 48 h to fix the tissue. The tissue was then dehydrated, embedded in paraffin, sectioned, and stained with HE staining to facilitate microscopic observation of the pathological changes in the mouse colon [47]. The colonic mucosa damage index (CMDI) score was calculated [48] and images were captured. Table 3 and Table 4 list these criteria.

### 4.7. Enzyme-Linked Immunosorbent Assay

The expression levels of IL-1β, IL-6, TNF-α, IL-4, IL-10, chemokine CXCL1, and CXCL2 in serum and colon tissue samples were detected using enzyme-linked immunosorbent assay (ELISA) kits (Shanghai Jianglai Industrial Co., Ltd., Shanghai, China). Absorbance was measured at a wavelength of 450 nm using a microplate reader and a standard curve was used for concentration conversion. The expression levels of MPO, ELANE, matrix metalloproteinase (MMP)-8, and MMP-9 in feces were also detected using ELISA kits (Shanghai Jianglai Industrial Co., Ltd., Shanghai, China).

### 4.8. Immunohistochemistry

Paraffin-embedded colon tissue samples were sectioned, dewaxed, rehydrated with gradient ethanol, and treated with sodium citrate buffer (10 mM, pH 6.0) for 20 min, followed by 3% hydrogen peroxide for 10 min to inactivate endogenous peroxidase. The sections were incubated with polyclonal antibodies recognizing MMP-8 (1:200; catalog number M09JA12; Chengdu Zhengneng Biotechnology Co., Ltd., Chengdu, China), MMP-9 (1:200; catalog number L05DE16; Chengdu Zhengneng Biotechnology Co., Ltd.), MPO (1:200; catalog number M05JA20; Chengdu Zhengneng Biotechnology Co., Ltd.), protein arginine deiminase (PAD)4 (1:5000; catalog number 00102732; Proteintech Group, Rosemont, IL, USA), ELANE (1:200; catalog number M05JA20; Chengdu Zhengneng Biotechnology Co., Ltd.), and ROS (1:200; catalog number 53q9220; Affinity Biosciences, Cincinnati, OH, USA) at 4 °C overnight. They were then incubated with the secondary antibody for 20 min, following which the color was developed with 3,3-diaminobenzidine. Sections were counterstained with hematoxylin, sealed with neutral resin, and observed under a light microscope (Olympus BX51; Fulai Optical Technology Co. Ltd., Shanghai, China). Image analysis was performed using Image J 1.53k software.

### 4.9. Immunofluorescence

Paraffin-embedded colon tissue samples were sectioned, dewaxed, rehydrated with gradient ethanol, and treated with sodium citrate buffer (10 mM, pH 6.0) for 20 min, followed by 3% hydrogen peroxide for 10 min to inactivate endogenous peroxidase. The sections were incubated with a rabbit anti-mouse Ly6G polyclonal antibody (1:500; catalog number M09JA12; Chengdu Zhengneng Biotechnology Co., Ltd.) overnight at 4 °C followed by a goat anti-rabbit IgG (Alexa Fluor 488) fluorescent secondary antibody (1:500; catalog number L08AU3A; Chengdu Zhengneng Biotechnology Co., Ltd.) for 20 min. The nucleus was stained with DAPI (Shandong Sijie Biotechnology Co., Ltd., Jinan, China) for 5 min, and sections were observed under a fluorescence microscope within 1 h.

### 4.10. Western Blotting

Protein was extracted from colon tissue using phenylmethylsulfonyl fluoride (catalog number 23184510; Biosharp, Hefei, China). Protein separation was performed using sodium dodecyl-sulfate polyacrylamide gel electrophoresis and proteins were transferred onto a polyvinylidene fluoride membrane using electroblotting. The membrane was blocked with 5% non-fat milk powder for 2 h and then the membranes were incubated with rabbit anti-mouse PAD4 (1:1000), rabbit anti-mouse citrullinated histone H3 (Cit H3; 1:1000; catalog number 00057440; Proteintech Group), rabbit anti-mouse MPO (1:5000), and rabbit anti-mouse β-actin (1:10,000; catalog number 380624; Chengdu Zhengneng Biotechnology Co., Ltd.) antibodies overnight at 4 °C. Subsequently, the membrane was incubated with secondary antibodies for 1 h and proteins were detected with enhanced chemiluminescence.

### 4.11. NET Visualization

Extracellular NETs were detected using immunofluorescence co-localization. Paraffin-embedded colon tissue samples were sectioned, dewaxed, rehydrated with gradient ethanol, and treated with sodium citrate buffer (10 mM, pH 6.0) for 20 min, followed by 3% hydrogen peroxide for 15 min in the dark to inactivate endogenous peroxidase. Then, sections were incubated with 3% BSA-PBS solution for 30 min at room temperature, followed by rabbit anti-mouse ELANE (1:200) and rabbit anti-mouse MPO (1:100) polyclonal antibodies overnight at 4 °C. A goat anti-mouse/rabbit HRP polymer from the goat anti-mouse/rabbit multiplex IHC detection kit (double) (Zen-Bio Inc., Durham, NC, USA) was then added to sections, which were incubated at room temperature for 1 h in the dark. After washing with PBS (pH 7.4), TSA-520 was added and sections were incubated at room temperature for 15 min. Antigen retrieval was repeated, and sections were then incubated with rabbit anti-mouse MPO (1:100) and rabbit anti-mouse Cit H3 (1:100) polyclonal antibodies overnight at 4 °C. A goat anti-mouse/rabbit HRP polymer was added and incubated at room temperature for 1 h in the dark. TSA-570 was added and sections were incubated at room temperature for 15 min. Sections were counterstained with DAPI, incubated in the dark at room temperature for 5 min, mounted, and observed under a fluorescence microscope within 1 h.

### 4.12. Statistical Analysis

All data were analyzed using SPSS software version 23.0 (IBM Corp., Armonk, NY, USA). Measurement data are expressed as mean ± standard deviation, and differences between groups were compared using one-way analysis of variance, with *p* < 0. 05 considered to indicate a statistically significant difference. The experiments were repeated three times.

## 5. Conclusions

The release of NETs by PMNs may be a major factor in the pathogenesis of UC. NETs are formed through a variety of factors and respond to a variety of stimuli, which makes it difficult for the single-component drugs currently used in clinical practice to play an effective role in the treatment of UC by regulating the release of NETs from PMNs. The findings suggest that BEPD regulates NET formation in PMNs through multiple targets and pathways to play a therapeutic role in UC.

## Figures and Tables

**Figure 1 pharmaceuticals-17-01077-f001:**
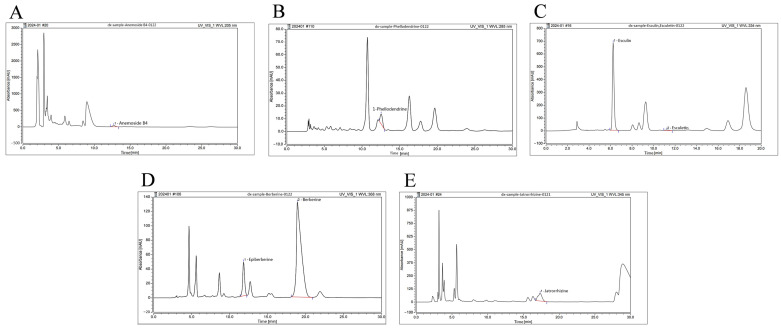
HPLC fingerprint of major components of BEPD. HPLC fingerprint of (**A**) anemoside B4, (**B**) phellodendrine, (**C**) esculin and esculetin, (**D**) epiberberine and berberine, and (**E**) jatrorrhizine of BEPD.

**Figure 2 pharmaceuticals-17-01077-f002:**
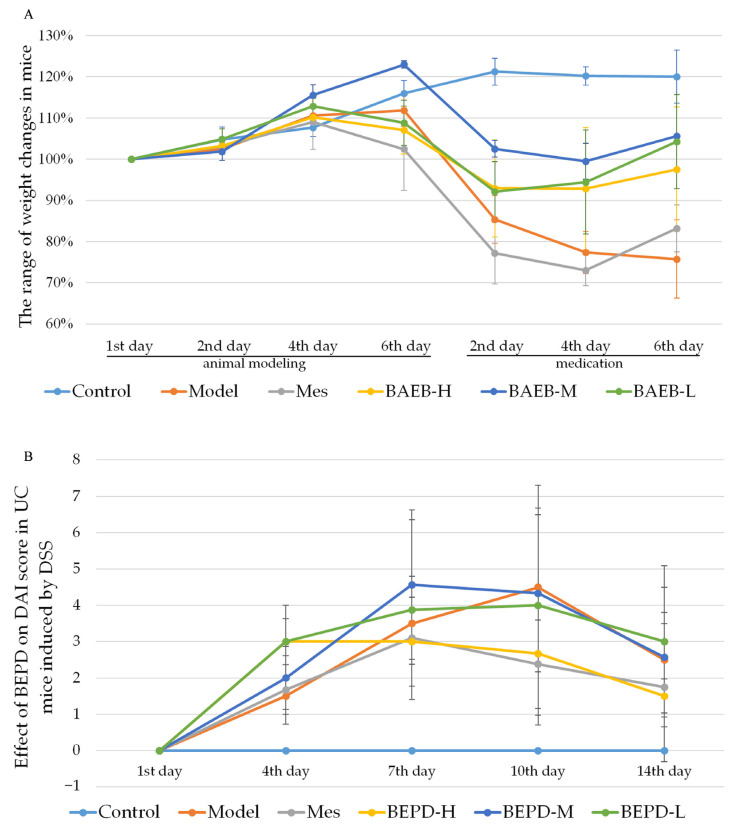
(**A**) Body weight changes. (**B**) Effect of BEPD on DAI score in UC mice induced by DSS.

**Figure 3 pharmaceuticals-17-01077-f003:**
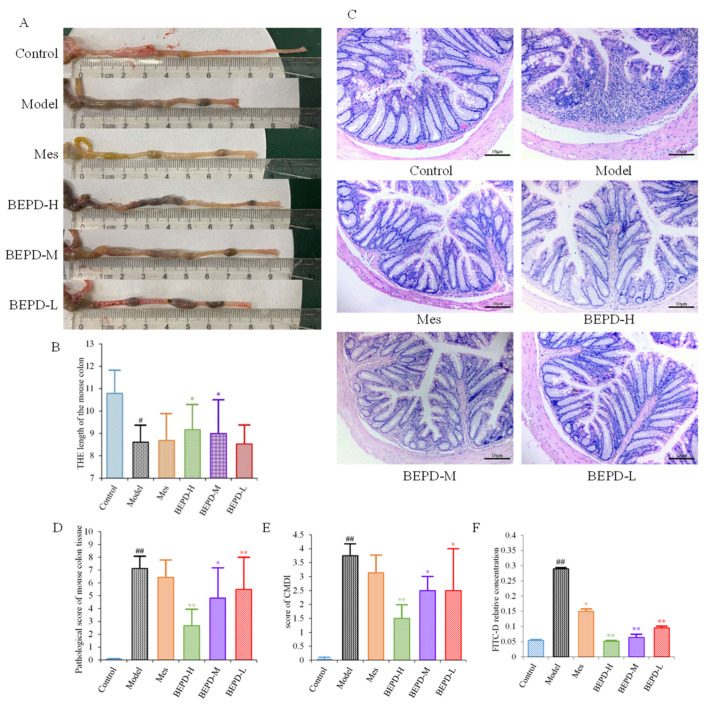
Anti-UC effects of BEPD against DSS-induced UC mice. (**A**,**B**) Representative pictures of colons from each group and the length of the mouse colon. (**C**) Representative HE staining of distal colon tissues (200× magnification). (**D**) Pathological score of mouse colon tissue. (**E**) Score of CMDI. (**F**) In the colon mucosal permeability experiment. (^#^ *p* < 0.05 vs. control group, ^##^ *p* < 0.01 vs. control group, * *p* < 0.05 vs. model group, ** *p* < 0.01 vs. model group. * *p* < 0.05 indicates statistically significant).

**Figure 4 pharmaceuticals-17-01077-f004:**
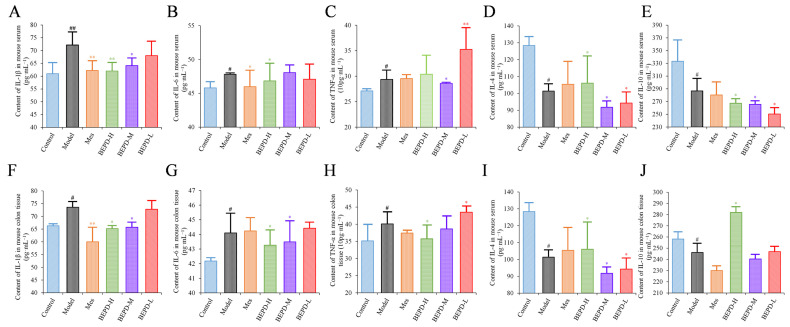
Effects of BEPD on cytokine production. (**A**–**C**) The levels of pro-inflammatory cytokines IL-1β, IL-6, and TNF-α in serum of UC mice. (**D**,**E**) The levels of anti-inflammatory cytokines IL-4 and IL-10 in serum of UC mice. (**F**–**H**) The levels of pro-inflammatory cytokines IL-1β, IL-6, and TNF-α in colon tissues of UC mice. (**I**,**J**) The levels of anti-inflammatory cytokines IL-4 and IL-10 in colon tissues of UC mice. (^#^ *p* < 0.05 vs. control group, ^##^ *p* < 0.01 vs. control group, * *p* < 0.05 vs. model group, ** *p* < 0.01 vs. model group. * *p* < 0.05 indicates statistically significant).

**Figure 5 pharmaceuticals-17-01077-f005:**
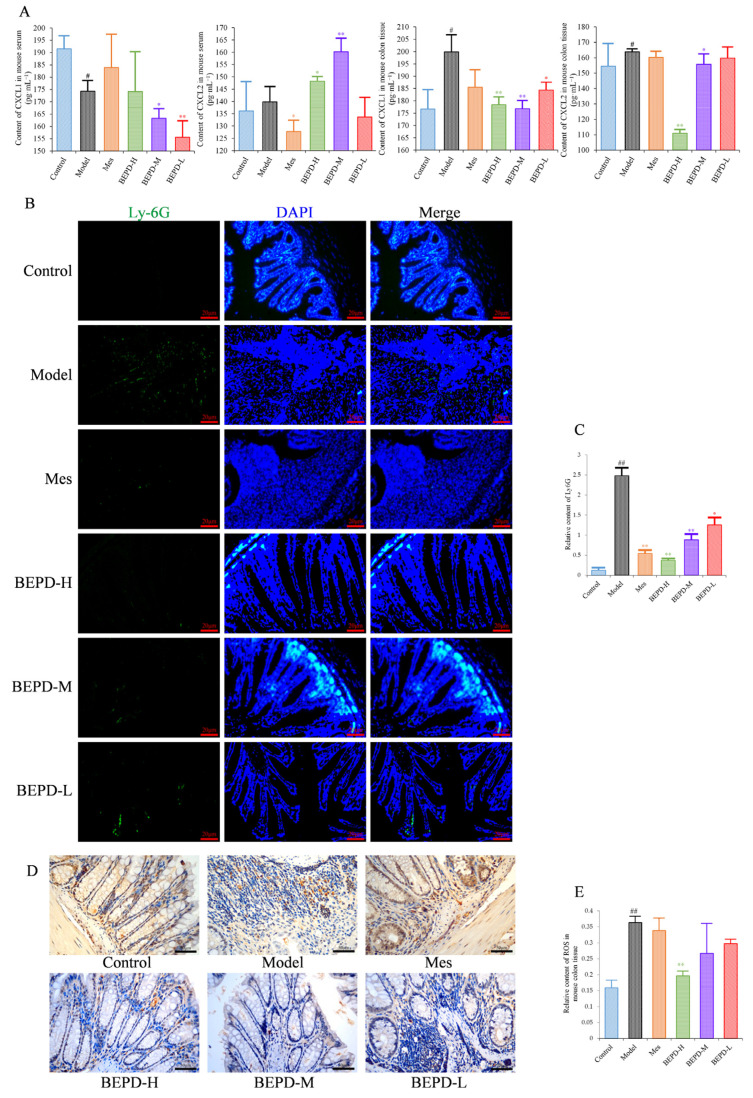
Effects of BEPD on chemokine production and PMN infiltration and activation. (**A**) The levels of neutrophil chemokines CXCL1 and CXCL2 in serum and colon tissues of UC mice. (**B**,**C**) Effects of BEPD on PMN infiltration in the colon mucosal tissues (400× magnification). (**D**) IHC staining of colon ROS of mice in each group (400× magnification). (**E**) Semi-quantitative results of IHC staining analysis for ROS. (^#^ *p* < 0.05 vs. control group, ^##^ *p* < 0.01 vs. control group, * *p* < 0.05 vs. model group, ** *p* < 0.01 vs. model group. * *p* < 0.05 indicates statistically significant).

**Figure 6 pharmaceuticals-17-01077-f006:**
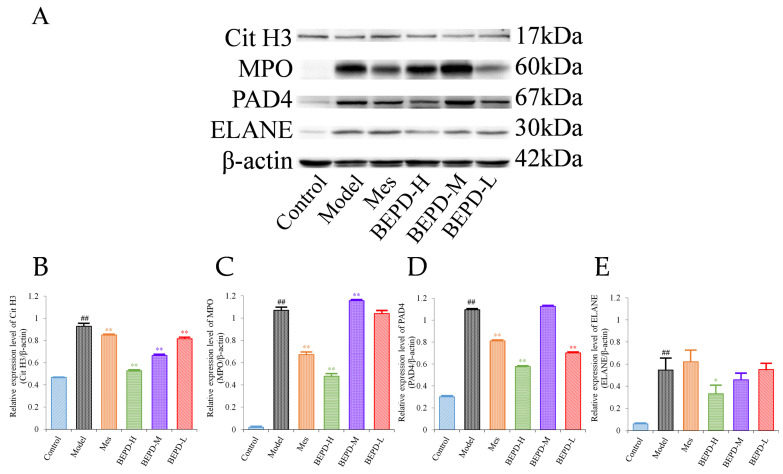
Effects of BEPD on the formation of NETs. (**A**–**E**) Western blot results of key proteins involved in NET formation, Cit H3, MPO, PAD4, and ELANE. (^##^ *p* < 0.01 vs. control group, * *p* < 0.05 vs. model group, ** *p* < 0.01 vs. model group. * *p* < 0.05 indicates statistically significant).

**Figure 7 pharmaceuticals-17-01077-f007:**
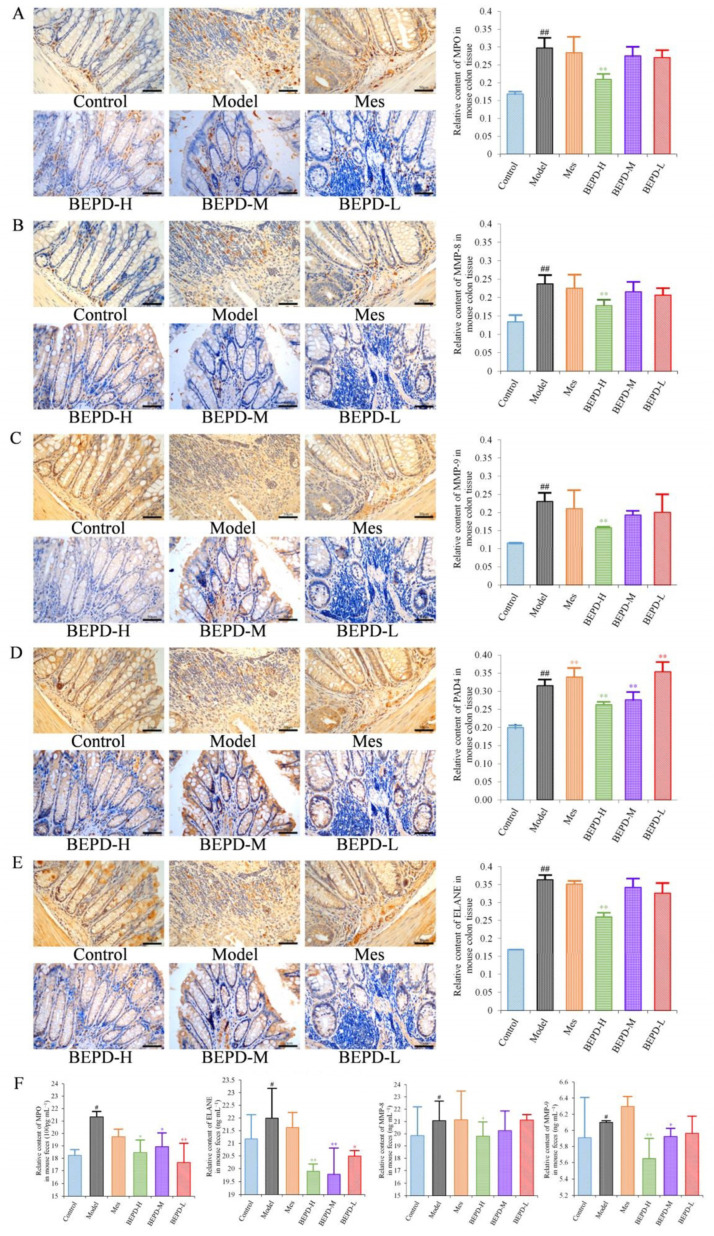
Effects of BEPD on the release of NETs. (**A**–**E**) IHC staining and semi-quantitative results of colon MPO, MMP-8, MMP-9, PAD4, and ELANE of mice in each group (400× magnification). (**F**) Relative content of MPO, MMP-8, MMP-9, and ELANE in mouse feces. (^#^ *p* < 0.05 vs. control group, ^##^ *p* < 0.01 vs. control group, * *p* < 0.05 vs. model group, ** *p* < 0.01 vs. model group. * *p* < 0.05 indicates statistically significant).

**Figure 8 pharmaceuticals-17-01077-f008:**
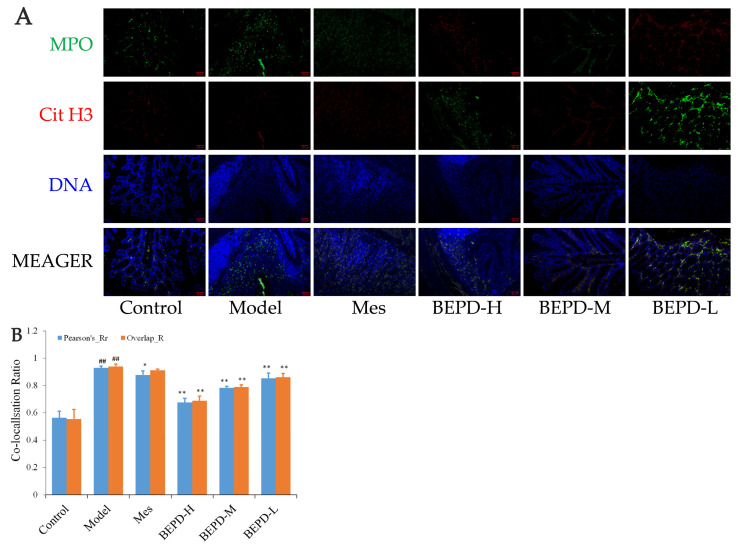
Effect of BEPD on extracellular NET. (**A**,**B**) MPO and Cit H3 co-located with DNA. (**C**,**D**) ELANE and MPO co-located with DNA. (^##^ *p* < 0.01 vs. control group, * *p* < 0.05 vs. model group, ** *p* < 0.01 vs. model group. * *p* < 0.05 indicates statistically significant).

**Table 1 pharmaceuticals-17-01077-t001:** The retention time, peak areas, and contents of esculin, esculetin, epiberberine, berberine, phellodendrine, jatrorrhizine, and anemoside B4 in BEPD.

Main Components	Retention Time (min)	Peak Area (mAU × min)	Content (μg/g)
Anemoside B4	12.711	17.506	78,116.529
Phellodendrine	12.530	2.828	3294.236
Berberine	18.913	99.167	65,110.351
Epiberberine	11.833	13.923	4177.739
Esculin	6.238	124.418	59,261.456
Esculetin	11.223	1.531	514.884
Jatrorrhizine	17.328	47.883	14,949.716

**Table 2 pharmaceuticals-17-01077-t002:** Scoring criteria of DAI.

Score	Weight Loss (%)	Stool Condition	Fecal Blood
0	*n* ≤ 1%	formed, moderate hardness	no abnormalities
1	1% < *n* ≤ 5%	formed, soft, not adhering to the perianal area	feces with dark red spots
2	5% < *n* ≤ 10%	soft and sticky around the anus	feces with dark red spots and visible bleeding around the anus
3	*n* > 10%	unformed, adhering to the perianal area, diarrhea	deep red feces with adhesion of perianal blood

**Table 3 pharmaceuticals-17-01077-t003:** Scoring criteria of colonic histopathology.

Score	Severity of Inflammation	Range of Inflammation	Amount of Crypt Damage
0	none	none	none
1	mild	mucosa	1/3 damaged
2	moderate	mucosa and submucosa	2/3 damaged, and epithelial surface present
3	severe	transmural	crypts and epithelial surface lost

**Table 4 pharmaceuticals-17-01077-t004:** Scoring criteria of CMDI.

Score	Mucosal Injury Degree
0	normal intestinal mucosa
1	mucosa congestion without ulcer lesions and bleeding
2	sporadic mucosal ulcer or slight bleeding
3	extensive ulcer necrosis or adhesion of intestinal mucosa and bleeding
4	severe bleeding and megacolon or stenosis or perforation

## Data Availability

Data is contained within the article.

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
