# Peer review of "Effects of the N-Butanol Extract of Pulsatilla Decoction on Neutrophils in a Mouse Model of Ulcerative Colitis"

_pharmaceuticals, 2024, doi:10.3390/ph17081077_

Round 1

Reviewer 1 Report

Comments and Suggestions for Authors

The study entitled "Effects of the n-butanol extract of Pulsatilla decoction on neutrophils in a mouse model of ulcerative colitis" aims to elucidate the molecular mechanisms underlying the effects of BEPD on UC. The existing literature extensively addresses this issue, with research indicating that the pharmacological mechanism likely involves maintaining gut microbiota homeostasis and diversity, increasing SCFA content, and repairing the colonic mucosal barrier (DOI: 10.1016/j.jep.2022.115741). Regarding the aim of investigating "neutrophil extracellular trap (NET) formation by neutrophils," a 2020 study assessed intestinal neutrophil infiltration into the intestinal wall (DOI: 10.21203/rs.3.rs-70028/v1).

In cancer cells, it has been found that the PI3K/AKT, TNF, and IL-17 pathways mediate the apoptotic effects of PD in HCC cells (DOI: 10.1186/s12906-023-04244-w), and that its bioactive component β-peltatin induces G2/M cell cycle arrest and apoptosis in pancreatic cancer (DOI: 10.1186/s13020-023-00774-0).

These findings, all from 2020-2023 examples, indicate that the conclusions of the present work have already been extensively published. Consequently, the novelty of the current study is low, rendering it below the quality standards expected by this journal.

Comments on the Quality of English Language

The English quality language is ok.

Author Response

Comments 1

The study entitled "Effects of the n-butanol extract of Pulsatilla decoction on neutrophils in a mouse model of ulcerative colitis" aims to elucidate the molecular mechanisms underlying the effects of BEPD on UC. The existing literature extensively addresses this issue, with research indicating that the pharmacological mechanism likely involves maintaining gut microbiota homeostasis and diversity, increasing SCFA content, and repairing the colonic mucosal barrier (DOI: 10.1016/j.jep.2022.115741). Regarding the aim of investigating "neutrophil extracellular trap (NET) formation by neutrophils," a 2020 study assessed intestinal neutrophil infiltration into the intestinal wall (DOI: 10.21203/rs.3.rs-70028/v1).

In cancer cells, it has been found that the PI3K/AKT, TNF, and IL-17 pathways mediate the apoptotic effects of PD in HCC cells (DOI: 10.1186/s12906-023-04244-w), and that its bioactive component β-peltatin induces G2/M cell cycle arrest and apoptosis in pancreatic cancer (DOI: 10.1186/s13020-023-00774-0).

These findings, all from 2020-2023 examples, indicate that the conclusions of the present work have already been extensively published. Consequently, the novelty of the current study is low, rendering it below the quality standards expected by this journal.

Response1: We greatly appreciate the reviewer's profound knowledge. Indeed, the pathogenesis of ulcerative colitis involves numerous factors, including disruption of gut microbiota, destruction of mucosal barriers, and so on. However, the key link is the immune inflammatory damage to the colon mucosa caused by these various factors. Neutrophils are an important component of the body's innate immunity, and existing research has shown that neutrophils and the NETs are involved in the occurrence and development of ulcerative colitis. Therefore, in this study, we aim to explore the mechanism of traditional Chinese medicine extract BEPD intervention in ulcerative colitis from the perspective of neutrophil regulation. Therefore, we have added the reasons for doing this research and the innovation points of this research, in the discussion section at the end of the manuscript. (page 11-12)

Comments on the Quality of English Language

The English quality language is ok.

Response2: Thank you! We have sent the manuscript to Editage for professional editing to emphasize this point. Editage is a brand of Cactus Communications (cactusglobal.com), a science communication and technology company. Since 2002, Editage has helped over 430.000 authors publish around 1.2 million research papers in scholarly journals across over 1000 disciplines through editorial, translation, transcription, and publication support services. (The editing certificate is attached at the back.)

Reviewer 2 Report

Comments and Suggestions for Authors

Dear authors,

I have some comments on your work:

1. Introduction needs a paragraph about this herbal formula. You consider that the reader already knows all about this, let say for aspirin of digoxin

2. last paragraph of introduction - a citation of your previous investigation is needed.

3. page 3, Subtitle 2.2. "DAI" needs explanation, it is the first time in the text - you give the explanation only in material & method section, at the end of the manuscript

4. Discussion is too short. It is rather a summary of your findings and nothing else. No comparison with the findings of others, no comments on the effects of different doses used. Should be improve

5. Section 4.2. 1st line. "BEPD was prepared as previously described [22]". BUT! the article is in Chinese!!! To my opinion, you consider that we, all of us, should already know about Chinese medicine and the Chinese language, as well - it's a bit disrespectful to the scientists of the western world. A short description is absolutely necessary. 

Author Response

Comments1. Introduction needs a paragraph about this herbal formula. You consider that the reader already knows all about this, let say for aspirin of digoxin

Response1: Thank you for pointing this out. We agree with this comment. Therefore, we have added  a paragraph(The fourth paragraph, marked in red) about this herbal formula in the part of introduction.

Comments2. last paragraph of introduction - a citation of your previous investigation is needed.

Response2: Thank you for pointing this out. We agree with this comment. Therefore, we have added  a citation of our previous investigation in last paragraph of introduction (added: We have previously demonstrated that BEPD exerts a therapeutic effect on UC by inhibiting PMN infiltration and the release of pro-inflammatory cytokines [25,26]). (page 3, lines 11)

Comments3. page 3, Subtitle 2.2. "DAI" needs explanation, it is the first time in the text - you give the explanation only in material & method section, at the end of the manuscript

Response3: Thank you for pointing this out. We agree with this comment. Therefore, we have added  a explanation of "DAI" at page 4,subtitle 2.2.( 2.2 BEPD significantly improves the general condition and disease activity index (DAI) scores of mice with UC)

Comments4. Discussion is too short. It is rather a summary of your findings and nothing else. No comparison with the findings of others, no comments on the effects of different doses used. Should be improve

Response4: Thank you for pointing this out. We agree with this comment. Therefore, in the first paragraph of the discussion, we have added a comparison with the results of other studies, adding a comment on the effects of using different doses. (page 12, Discussion)

Comments5. Section 4.2. 1st line. "BEPD was prepared as previously described [22]". BUT! the article is in Chinese!!! To my opinion, you consider that we, all of us, should already know about Chinese medicine and the Chinese language, as well - it's a bit disrespectful to the scientists of the western world. A short description is absolutely necessary. 

Response5: Thank you very much for your kind criticism and very insightful opinions. Therefore,  while we have retained the references, a short description have added. (page 13, Section 4.2.)

In brief, Radix pulsatillae, Cortex phellodendri, Rhizoma coptidis and Cortex fraxini were mixed at a ratio of 15:12:6:12, soaked in 80% ethanol, and heated in a 70°C water bath overnight to extract the Pulsatilla decoction, refluxed evaporation for 3h and filter after cooling. The process was repeated three times. The filtrate was combined, and then extracted with petroleum ether, ethyl acetate, and n-butanol to extract different effective parts separately. The extract was dried and evaporated at 80°C by rotary evaporator, and finally, the BEPD was obtained.

Reviewer 3 Report

Comments and Suggestions for Authors

The Authors used in the analytical part of the study the HPLC method. Some information is lacking:

1. What was the flow of the mobile phase for the analysis?

2. Was the HPLC method developed or was it was previously published?

4. If the method was developed, was it optimized?

5. Was the method validated? If so, according to what criteria?

6. The number of the decimal places should be unified (table 4).

7. For retention time of Esculetin there is a missing decimal separator - 11223. In my opinion it should be 11.223 (Table 4).

8. The quality of Figure 1 is poor, as well as the description. What present the chromatograms from A to E?

9. In chromatogram E in figure 1 there is big signal at the and of the analysis. What is it? The analysis was finished before the analyte was eluted.

Author Response

Comments 1. What was the flow of the mobile phase for the analysis?

Response: Thank you for pointing this out. We agree with this comment. Therefore, we have added the flow of the HPLC method. (page 14, lines 7-21)

Anemoside B4, phellodendrine, berberine, jatrorrhizine, esculin, esculetin and epiberberine contents in BEPD were detected via HPLC. A sample of BEPD powder weighing 0.2 g was precisely weighed and passed through No.2 sieve (850μm±29μm 24mesh) . And placed in a conical flask with stopper and a mixed solution of methanol-hydrochloric acid (100:1) totaling 50 mL was accurately added. The bottle was sealed, weighed, and subjected to ultrasonic treatment (Power: 250 W, Frequency: 40 kHz) for 30 minutes. The bottle was then cooled, reweighed, and made up to the original weight using methanol. The resulting mixture was shaken well, filtered, and 2 mL of the filtrate was precisely measured and placed in a 10 mL volumetric flask. Add methanol into the volumetric flask to calibration, shaken well, filtered, took the filtrate. Liquid chromatography conditions: Syncronic C18, Dim 250 mm × 4.6 mm, 5 μm chromatographic column was used. Octadecyl silane bonded silica as filler, with acetonitrile (0.05 mol/L):potassium dihydrogen phosphate solution (50:50) (Add 0.4g sodium dodecyl sulfate to each 100ml mobile phase, and adjust the pH value to 4.0 with phosphoric acid) as mobile phase. The evaluation was conducted using an external standard quantitative method.

Comments 2. Was the HPLC method developed or was it was previously published?

Response: Thank you for pointing this out. We agree with this comment. The analysis part of this study was entrusted to Zhongjia Testing (Guangzhou) Co., Ltd. for testing, the technology is mature, and our research team has entrusted them for drug composition analysis for many times. There is a technical service contract attached.

Comments 4. If the method was developed, was it optimized?

Response: Thank you for pointing this out. We agree with this comment. The analysis part of this study was entrusted to Zhongjia Testing (Guangzhou) Co., LTD., who partially optimized the operation process. For example, the mobile phase was further modified by adding 0.4 g of sodium dodecyl sulfate to every 100 mL, and the pH was adjusted to 4.0 using phosphoric acid.

Comments 5. Was the method validated? If so, according to what criteria?

Response: Thank you for pointing this out. We agree with this comment. In this method, the standard substance of anemoside B4, phellodendrine, berberine, jatrorrhizine, esculin, esculetin and epiberberine were used as the reference, and the external standard quantitative method was used to evaluate the component content of the sample.

Comments 6. The number of the decimal places should be unified (table 4).

Response: We are truly grateful for the help of reviewer. We have corrected the errors in the revised manuscript. Thanks again for your valuable suggestions. (table 4)

Comments 7. For retention time of Esculetin there is a missing decimal separator - 11223. In my opinion it should be 11.223 (Table 4).

Response: We are truly grateful for the help of reviewer. We have corrected the errors in the revised manuscript. Thanks again for your valuable suggestions. (table 4).

Comments 8. The quality of Figure 1 is poor, as well as the description. What present the chromatograms from A to E?

Response: Thank you for pointing this out. We agree with this comment. Therefore, we have replaced high-quality pictures and chromatograms from A to E were redescribed.

Comments 9. In chromatogram E in figure 1 there is big signal at the and of the analysis. What is it? The analysis was finished before the analyte was eluted.

Response: Thank you for pointing this out. We agree with this comment. Therefore, we have consulted the technical personnel of Zhongjia Testing (Guangzhou) Co., LTD., and their reply was that "we do not do research on substances other than test items, we only look at substances corresponding to standard materials". Therefore, we judge that the question you raised may not have an impact on the results of the analysis.

Reviewer 4 Report

Comments and Suggestions for Authors

After reading the article Effects of the n-butanol extract of Pulsatilla decoction on neutrophils in a mouse model of ulcerative colitis, where the authors want to explore the molecular mechanism underlying the effects of BEPD on UC, in particular its effects on neutrophil extracellular trap (NET) formation by neutrophils, the results seemed very interesting to me with the option of publication. However, I have several suggestions:

1. The authors must attach information about BEPD and its relationship with colitis in the introduction.

2. What was the BEPD dissolved in and in what volume was it administered?

3. Better the quality of the figures.

4. I suggest the authors discuss the results obtained based on the active principles they found from BEPD.

5. In the references, put the year in bold and the title of the magazine in italics.

Comments on the Quality of English Language

Minor editing of English language required

Author Response

Comments 1. The authors must attach information about BEPD and its relationship with colitis in the introduction.

Response 1: Thank you for pointing this out. We agree with this comment. Therefore, we have attached information about BEPD and its relationship with colitis in the introduction. (page 2)

Comments 2. What was the BEPD dissolved in and in what volume was it administered?

Response 2: Thank you for pointing this out. We agree with this comment. Therefore, we have added information about BEPD. The dissolved substance of BEPD was normal saline. According to the results of the preliminary experiment, the doses of 80mg/kg, 40mg/kg and 20mg/kg were selected for the subsequent experiment. (page 14, section 4.3)

Comments 3. Better the quality of the figures.

Response 3: Thank you for pointing this out. We agree with this comment. Therefore, we have replaced the figures with better quality.

Comments 4. I suggest the authors discuss the results obtained based on the active principles they found from BEPD.

Response 4: Thank you for pointing this out. We agree with this comment. Therefore, we have re-edited the discussion based on the active principles we found from BEPD. (page 11-13)

Comments 5. In the references, put the year in bold and the title of the magazine in italics.

Response 5: We are truly grateful for the help of reviewer. We have corrected the errors in the references. Thanks again for your valuable suggestions.

Comments on the Quality of English Language

Minor editing of English language required

Response 6: Thank you for pointing this out. We agree with this comment. Therefore, we have sent the manuscript to Editage for professional editing to emphasize this point. Editage is a brand of Cactus Communications (cactusglobal.com), a science communication and technology company. Since 2002, Editage has helped over 430.000 authors publish around 1.2 million research papers in scholarly journals across over 1000 disciplines through editorial, translation, transcription, and publication support services. (The editing certificate is attached at the back.)

Round 2

Reviewer 1 Report

Comments and Suggestions for Authors

The authors improved the manuscript.

Please, fix all the manuscript avoiding the use of "we", "our", etc. Science is written in the third person.

Author Response

Comments 1: Please, fix all the manuscript avoiding the use of "we", "our", etc. Science is written in the third person.

Response 1: The opinions of reviewers are indeed very important. We have made revisions according to the feedback, and the changes have been highlighted in yellow. Thank you!

Reviewer 3 Report

Comments and Suggestions for Authors

I still have one question: What was the flow of the mobile phase? I can't find this information. I appreciate that Authors improved the description of HPLC analysis, but the information about the mobile phase flow is still missing. Was it 1 ml/min?

Author Response

Comments 1: I still have one question: What was the flow of the mobile phase? I can't find this information. I appreciate that Authors improved the description of HPLC analysis, but the information about the mobile phase flow is still missing. Was it 1 ml/min?

Response 1: Thank you very much for the valuable feedback from the reviewers. Due to the length of the article, the description in this section is not detailed enough. According to the principle of HPLC, when the components dissolved in the mobile phase pass through the stationary phase, they flow out of the stationary phase successively due to the different sizes and strengths of their interactions (adsorption, distribution, exclusion, affinity) with the stationary phase, resulting in different retention times in the stationary phase. In the HPLC detection process of this study, the mobile phase consisted of acetonitrile and potassium dihydrogen phosphate solution, and the stationary phase was a Syncronic C18 chromatographic column. The flow rate was 1 mL/min.